# Developing a context-relevant psychosocial stimulation intervention to promote cognitive development of children with severe acute malnutrition in Mwanza, Tanzania

**Cecilie L. Jensen**[1⊙], **Erica Sanga**[2⊙], **Heather Kitt**[3], **George PrayGod**[2], **Happiness Kunzi**[2], **Theresia Setebe**[2], **Suzanne Filteau**[4], **Jayne Webster**[5], **Melissa Gladstone**[3⊙], **Mette F. Olsen** [1,6] *

1 Department of Nutrition, Exercise and Sports, University of Copenhagen, Copenhagen, Denmark, 2 National Institute for Medical Research, Mwanza Centre, Mwanza, Tanzania, 3 Department of Women and Children's Health, Institute of Life Course and Medical Sciences, University of Liverpool, Liverpool, United Kingdom, 4 Faculty of Epidemiology and Population Health, London School of Hygiene and Tropical Medicine, London, United Kingdom, 5 Faculty of Infectious and Tropical Diseases, London School of Hygiene and Tropical Medicine, London, United Kingdom, 6 Department of Infectious Diseases, Rigshospitalet, Copenhagen, Denmark

⊙ These authors contributed equally to this work.
* mette.frahm.olsen@regionh.dk

**Data Availability Statement:** The Medical Research Coordinating Committee (MRCC) of the National Institute for Medical Re-search (NIMR) in

## Abstract

More than 250 million children will not meet their developmental potential due to poverty and malnutrition. Psychosocial stimulation has shown promising effects for improving development in children exposed to severe acute malnutrition (SAM) but programs are rarely implemented. In this study, we used qualitative methods to inform the development of a psychosocial stimulation programme to be integrated with SAM treatment in Mwanza, Tanzania. We conducted in-depth interviews with seven caregivers of children recently treated for SAM and nine professionals in early child development. We used thematic content analysis and group feedback sessions and organised our results within the Nurturing Care Framework. Common barriers to stimulate child development included financial and food insecurity, competing time demands, low awareness about importance of responsive caregiving and stimulating environment, poor father involvement, and gender inequality. Caregivers and professionals suggested that community-based support after SAM treatment and counselling on psychosocial stimulation would be helpful, e.g., how to create home-made toys and stimulate through involvement in everyday chores. Based on the findings of this study we developed a context-relevant psychosocial stimulation programme. Some issues identified were structural highlighting the need for programmes to be linked with broader supportive initiatives.

Tanzania demand that data should not be shared publicly or transferred without their permission. Data are available upon request for researchers who meet the criteria for access to confidential data. They may use the contact details below to request the data: The Secretariat, Na-tional Institute for Medical Research, 2448, Barack Obama Road, P O Box 9653 Dar Es Salaam, Tanzania. E-mail: zt. ro.rmin@scihte.

**Funding:** The study was funded by a joint global health trial development grant (MR/T003731/1) from the Department of Health and Social Care (DHSC), the Foreign, Commonwealth & Development Office (FCDO), the UK Medical Research Council (MRC) and Wellcome. The funders had no role in study design, data collection and analysis, decision to publish, or preparation of the manuscript.

**Competing interests:** The authors have declared that no competing interests exist.

# Introduction

It has been estimated that more than 250 million children below 5 years do not reach their full developmental potential, mainly due to poverty and malnutrition [1]. Most of these children live in low- and middle-income countries with the highest prevalence found in sub-Saharan Africa [2].

Early childhood is a critical period for brain development with most brain growth happening in the first five years. Factors that affect brain growth in these early years may therefore affect children's development and cognition. Malnutrition is one of those factors where we know that children exposed to malnutrition in early life have poorer school performance and lower income in adult life than their peers [3–5].

Children are defined as having severe acute malnutrition (SAM) if they have a mid-upper arm circumference <115 mm, a weight-for-height z-score <-3 as per WHO growth standards [6] or bilateral pitting oedema. It is well-known that SAM is associated with impaired cognitive and motor development [7–9], but deficits in child development are not only a consequence of inadequate diets. Families exposed to malnutrition are often also affected by psychological distress and children are likely to be offered little stimulation and responsive care [10]. Previous research has highlighted the need for providing family support focusing on responsive caregiving and psychosocial stimulation to support the development of the child after an episode of malnutrition [11–13]. An early study from Jamaica in the 1970s showed notable effects of an intense stimulation intervention delivered through daily activities during admission for severe malnutrition and weekly home visits following discharge [14]. These findings led to the inclusion of psychosocial stimulation as one of the 10 steps in the WHO guidelines for hospital-based SAM management [15]. This step includes guidance on; tender loving care, a cheerful and stimulating environment, structured play activities for 15–30 minutes per day, support for the maternal environment etc. However, structures are not in place to enable this type of intervention within most health services in low- and middle-income settings.

In general, psychosocial stimulation refers to activities and interventions that aim to enhance an individual's psychological and social well-being. This can include cognitive, emotional, and social aspects of development. Since the first studies in the 1970s, more recent interventions for promoting responsive care have shown some effects, but often not when implemented at scale or within systems of care [16–25]. There is still little knowledge of the pathways of intervention components, how they are modified by contextual factors and how they can be scaled [12, 26–28]. In practice, support for stimulation and responsive caregiving is rarely offered during hospital-based treatment, and it is still not mentioned in the guidelines for community-based treatment of SAM [29].

The lack of psychosocial stimulation activities implemented in SAM management is likely due to resource-constraints, such as training of staff and supervision [30]. It is therefore imperative that interventions are created that are effective, context-relevant and can be implemented at scale in existing health services in low- and middle-income countries. We still do not have good evidence as to what factors may facilitate effective implementation of psychosocial stimulation programmes for this vulnerable group [23, 31]. To better understand strategies for implementation and facilitate scale-up, it is crucial to gain the insight of caregivers and professionals who work with SAM and early child development [32, 33]. In this study in Mwanza, Tanzania, we therefor aimed 1) to explore barriers and facilitators in promoting psychosocial stimulation of children treated for SAM, and 2) to explore perceptions of early child development among caregivers and professionals and their suggestions as how best to design a context-specific psychosocial stimulation package to be integrated with SAM management.

## Materials and methods

### Study design and setting

With a long-term plan of conducting a factorial trial to test the effects of improved nutritional and psychosocial stimulation interventions in the management of malnutrition, we conducted the 'BrightSAM (Brain development, growth and health in children with SAM) trial development study' between June 2020 –February 2022. The overall aim was to develop, pilot test and assess acceptability and implementation feasibility of the nutrition and psychosocial stimulation interventions to be integrated with SAM management. Additional details of the trial development study have been described elsewhere [34]. In the present paper, we report findings of qualitative formative research we conducted to develop the content and delivery mode of the stimulation intervention. Our intention was to understand how we could create an intervention feasible for use in both hospital and community management of SAM. The findings of this formative research informed the development of a psychosocial stimulation program that was integrated with SAM management in a pilot study at Bugando Medical Centre (BMC). The results on impact on child development outcomes and on implementation fidelity and feasibility will be reported separately.

The study was conducted at the National Institute for Medical Research (NIMR) in collaboration with BMC and the Tanzania Home Economics Association (TAHEA) in Mwanza, Tanzania. Mwanza is located on the shore of Lake Victoria with a population of about 1,182,000 in 2021 [35]. Main economic activities in the region are agriculture, livestock keeping and fishing. The prevalence of stunting in Mwanza region is 39%, just above the country average of 34%. Among women in the region, 46% have completed primary school, 23% attended or completed secondary school or higher education, and female literacy rates are approximately 77% [36]. The nutrition unit at the BMC paediatric department manages cases of SAM with complications. The unit provides specialized care and feeding by nurses and a paediatrician. Staff also includes a nutritionist, a social worker and a physiotherapist. Conditions such as sickle cell disease, cerebral palsy and congenital heart diseases are common among malnourished children [37]. Outpatient follow-up clinics for malnutrition cases are supervised by a paediatrician, a nutritionist and a nurse.

This qualitative interpretivist study used in-depth interviews to explore the 'subjective interpreted' reality of parents/caregivers of SAM children, and professionals providing SAM services as they interacted within their worlds during their experience of caring for malnourished children [38]. We chose the interpretivist approach to understand the personal perceptions and experiences of participants on malnutrition and psychosocial stimulation in children, as well as motivations towards a potential intervention, based on the beliefs and norms of their communities and workplace. We felt that both caregivers, and professionals working in SAM units would have a range of experiences, views, and perspectives, based on their unique contexts, and their interactions with their communities and the services. In-depth interviews were our data collection method of choice as we wanted deep, unique insights from participants that was gathered one-on-one, with sensitivity, in a private space [39]. Exploring the views and experiences of a range of caregivers and professionals was expected to be valuable in providing insight into the different 'realities' that exist and the different ways that caregivers and professionals experience malnutrition and psychosocial stimulation in children (i.e. rich in information and lived the experience) [39, 40].

### Theoretical framework

As a theoretical framework for organizing and understanding our findings, we chose the Nurturing Care Framework which has been introduced by WHO and UNICEF to provide a

comprehensive approach to promote the health and well-being of young children [41]. The framework includes five components of relevance to early child development: 'good health', 'adequate nutrition', 'responsive caregiving', 'opportunities for early learning', and 'security and safety'. The framework aims to guide policies and practices that support optimal development and was therefore selected as we aim to use the results of this work pragmatically to consider how to create an intervention. We applied the framework to structure our findings, since the barriers and facilitators that were expressed by caregivers and professionals during interviews linked with one or more of the framework's components.

## Participant selection and data collection

Caregivers of children, who had recently been admitted with SAM and were receiving outpatient follow-up, were selected by purposive sampling [42] and invited for in-depth interviews. Healthcare staff from the nutrition unit at BMC assisted in identifying potential participants but were not involved in any communication about the study with caregivers to avoid introduction of response bias. Caregivers were contacted via phone and invited to BMC or NIMR for interviews. The number of interviews with caregivers was based on an assessment of data saturation, which was identified through preliminary analysis of data for emergence of new themes. The senior social scientist conducting interviews (ES) made the decision to stop further interviews, when the responses included only existing themes [43, 44]. Additional interviews were conducted with professionals.

Additional in-depth interviews were conducted with professionals who were purposively selected based on their experience and involvement with SAM children or projects related to malnutrition and/or early child development. The number of interviews with professionals was therefore based on an ambition to include all categories of staff with relevant experience of treatment of severe acute malnutrition and/or promotion of child development in the geographical area. The selected professionals were health care staff, who included 1 paediatrician, 1 nutritionist and 2 nurses from the BMC nutrition unit, and key staff from NGOs and projects working to promote child development in the community, including 4 from TAHEA engaged with promoting early child development in the community, and 1 from the Engaging Fathers for Effective Child Nutrition and Development in Tanzania (EFFECTs project). The interviews with professionals were conducted at their workplaces or community centres.

Interview guides were developed to explore awareness and attitudes towards early child development and psychosocial stimulation. The guides included questions about contextual factors that may be barriers or facilitators to support the learning and development of a child, as well as current support for early child development in Mwanza. The guides focused mainly on barriers and facilitators for supporting early child development at the family level, as the study was intended to inform development of an intervention delivered to families while in contact with health care teams during management of SAM. However, if barriers or facilitators at other levels came up during interviews, we probed further into these. Finally, we asked questions to gain specific views and suggestions on how a short, focused psychosocial stimulation intervention could be designed to improve cognitive development of children during treatment for SAM. Interview guides were pretested through four interviews; two with professionals and two with caregivers who were not otherwise part of the study. Subsequently, the interview guides were revised to expand probes for deeper exploration as needed and some questions paraphrased for clarity.

Data collection was done by experienced social science staff from NIMR. Data triangulation was achieved by interviewing both caregivers of children treated for SAM and professionals working with SAM and early child development. All interviews were conducted between June-

September 2020 and took place in quiet, private environments. They were conducted in Kiswahili by three trained, experienced research scientists from NIMR and lasted 45–60 minutes.

After preliminary data analysis, respondents were invited to a feedback session involving further discussions where they were invited to clarify or elaborate on views and suggestions from the interviews. Separate group discussions were conducted for the caregivers and the professionals. The sessions lasted 60–90 minutes and were both hosted at NIMR. These sessions were not intended to add new data for analysis but aimed to validate data obtained from the in-depth interviews to ensure correct understanding and to feed-back study findings to participants [45].

### Data management and analysis

The interviews were digitally recorded, transcribed *verbatim* in Kiswahili and translated into English by research assistants at NIMR fluent in both Kiswahili and English. Transcriptions and translations were cross-checked and verified by two authors (ES and GP), fluent in both languages. Transcripts were coded following principles of thematic content analysis [46] using Dedoose software (version 8.3.35, SocioCultural Research Consultants, Los Angeles, US). First, an initial analytical coding framework was developed deductively by the research team based on previous literature, assumptions and the interview guide. All codes were structured into a codebook with descriptions for each. Then, the first transcript was coded independently by each of the three coders (ES, CLJ, HK) and then cross-checked and reviewed by the team to ensure that a similar conceptualisation of framework and codes was used by all coders. Once this preliminary coding was completed, the primary coding framework was created. New codes were created inductively and added to the framework during the following analyses and data familiarisation. Multiple codes were applied to a section of text when appropriate. The remaining transcripts were coded by the coding team and verified by JW, MG and MFO.

Identified barriers and facilitators which emerged from the thematic content analysis were then organised and synthesised using the Nurturing Care Framework [38]. Based on this analysis, we report suggestions and opportunities for a psychosocial stimulation intervention to improve nurturing care and early child development among children recovering from SAM.

### Ethical considerations

The study received ethical clearance from the Medical Research Coordinating Committee of the National Institute for Medical Research—Dar es Salaam, Tanzania (Ref- NIMR/HQ/R.8c/ Vol.I/1708), and from the ethics committee of the London School of Hygiene and Tropical Medicine (Ref: 17831–1). Written informed consent was obtained prior to starting interviews with all caregivers and professionals. All quotes are used anonymously and without descriptive information that could lead to identification of respondents.

### Inclusivity in global research

The research involved both north and south partners including NIMR and BMC based in Mwanza. To further ensure that we took on board local experiences we invited TAHEA which is a locally based NGO working at grassroot level which has very strong links with the study population. The inclusion of TAHEA helped to refine our plans for data collection as well as interpretation of data and has increased the relevancy of the data generated to the local setting.

## Results

### Interviewee characteristics

A total of sixteen interviews were conducted. Seven included caregivers (all mothers) of children recently treated for SAM. The majority of caregivers had completed primary or secondary school and were working as small-scale farmers or vendors. The children were between 6 months and 5 years of age, and about half were girls. Most families lived within 10 km from BMC. As described above, nine interviews were conducted with local professionals working at the BMC nutrition unit or at an NGO involved in supporting early child development. All professionals had more than five years of work experience in their present field. Two caregivers declined participation in the study, one of them due to long distance to the place of interview and the other due to lack of childcare for older children in the household. None of the professionals invited for interviews declined.

### Barriers and facilitators for providing psychosocial stimulation

**Good health: The fundamentals of child development.**   Caregivers and professionals described how illness negatively impacts on children's growth and ability to learn and should therefore be addressed as a primary focus in any programme to support child development. One mother described how she could see the impact of illness on her child's development, with similar messages emerging from professionals.

*"Before falling into sickness, she was so active. She was able to walk around holding objects like tables, chairs. But after falling into sickness, everything stopped (. . .) sickness may slow down the normal growth of a child"*

(Mother 7).

*"If a child is frequently under infections (. . .) he will be late to walk, communicate and talk"*

(Professional 2).

Professionals described how increased support of children's health after discharge from the nutrition unit would be beneficial, in particular by strengthening existing links between the nutrition unit and the community health workers.

*"Community health workers should be trained and become knowledgeable, because us from here* (BMC) *we cannot go there always, we need them, they should be assisting in follow-up"*

(Professional 2).

**Ensuring adequate nutrition and resources at home.**   Inadequate nutrition was clearly perceived by caregivers as a vital barrier for learning and development of their children.

*"If the baby doesn't eat well–probably he cannot grow well, most of these children they are not active (. . .) because even for an adult who doesn't eat, he or she cannot be active–what about a little child (. . .) eating well is important for child development"*

(Mother 7).

Professionals also mentioned adequate nutrition when asked about the foundation for growth and development.

*"Good nutrition, because he can have all good environment and he is well stimulated, but that won't be enough if he doesn't have services like nutrition"*

(Professional 5).

Poverty was clearly perceived by both caregivers and professionals as a main barrier for the availability of nutritious food–and as such, a fundamental barrier that cannot easily be addressed by counselling or behaviour change interventions. Mothers demonstrated good awareness of the importance of nutrition and thus experienced the agony of knowing how to feed their child, but not having the resources to do so.

*"You can think of buying the required food for the child, but you find yourself with no money to buy* (. . .) *you love your children and wish to buy them everything good for their development, but due to the poor economic status you have no choice"*

(Mother 1).

Professionals described the poor socio-economic and educational status of the families of children with SAM and emphasized the importance of cooking demonstrations on how to prepare nutritious low-cost meals with locally available foods.

*"It's like a mind-set where one knows that to eat well it has to be expensive,* (. . .) *we are telling them to bring foods that can be found within their environment and we use those foods to cook a meal that contain five groups of nutrients,* (. . .) *so after you have cooked and prepared a balanced meal, they will say ooh so it's possible"*

(Professional 7).

However, one mother highlighted how the time demand for her income-generating activities resulted in little time for her to keep an eye on her child's diet, even to a point that led her child to become malnourished.

*"I was busy with work, actually that is what costed me, because the child was most of the time with the house girl, I had no time to check her diet, then she fell sick, then I was told it was malnutrition. I had to stop my work but it was too late"*

(Mother 5).

**Enabling responsive caregiving through the wider circle of care.** Professionals consistently described the importance of having multiple caregivers to support the child and family. This was seen as pivotal for good child-caregiver interactions.

*"When you find a family which is extended—I mean maybe if there is father, uncle, and others, there is grandmother, those children are so different, and their learning and development is very good because they have people to interact and play with"*

(Professional 4).

However, some professionals expressed that other caregivers (siblings or housekeepers), who take care of children due to the competing demands on the mother, may not be adequately able to provide responsive caregiving.

*"The mother is busy, a child is raised with a house girl. If the house girl is quiet, a child will learn to be quiet, if she does not get the opportunity to play with other people who can stimulate her"*

(Professional 2).

Gender inequities and expectations of gender roles was raised by interviewees as a barrier for responsive caregiving. It was highlighted by both caregivers and professionals that their society expects mothers to take care of the children whereas fathers are expected to bring money for the household. It emerged from our interviews, that these societal norms may lead to lack of involvement of fathers as they are expected to have other responsibilities and spend little time caregiving.

*"A father goes to work at 6 am and return home at 11 at night, what time will he play with his child? A child has already gone to bed, he doesn't have time"*

(Mother 2).

Some professionals highlighted their desire to involve fathers and suggested ways to reach fathers through community programmes.

*"When you go home you will find mothers alone and fathers are not around. So we thought, how can we access fathers (. . .). They are counselled where they are working or using their time (. . .), our counsellors are reporting that they are getting support from men, they are allowed for like twenty minutes while people are taking their coffee while listening to talks about care for child development"*

(Professional 5).

Other professionals primarily blamed lack of awareness for insufficient responsive caregiving.

*"When it comes to playing with a child and understanding a child's feelings, I don't think social economics plays a role, I think it's only knowledge (. . .) To smile for a child doesn't need money"*

(Professional 7).

**Awareness and prioritisation of play and opportunities for early learning.** Professionals and caregivers that we interviewed highlighted how crucial it is to provide young children opportunities to interact and stimulate their senses.

*"I teach him different things. I play with him, when I get time, I really try my best, even though household chores are all on me"*

(Mother 4).

However, professionals also expressed how a lack of awareness of the importance of play leads to lack of prioritisation. A professional described how some caregivers might prioritise cleanliness over social stimulation. Similarly, a mother highlighted lack of understanding as a barrier to psychosocial stimulation.

*"Some kids, their parents stop them from playing with other children, this child then stops playing with others. Parents might think that if the child plays with others, they become dirty*

*not knowing their children's cognitive development remains stunted as children learn mostly through games"*

(Professional 8).

*"I think education matters. Some mothers have not gone to school, she just gave birth to a child—she doesn't know about playing with a child, maybe she thinks it's waste of time. I think education should be expanded even in our community"*

(Mother 3).

Clearly, time constraints and competing responsibilities may act as barriers towards stimulating the child through play and interaction.

*"There are some mothers who are vendors, they leave home at 6 am with bucket of tomatoes. She spends a whole day at the market selling tomatoes, when will she play with a child? I have time because I'm a housewife"*

(Mother 6).

Other competing concerns among caregivers may also pose a barrier for them to provide responsive care. One professional highlighted how keeping the caregivers' wellbeing in mind is fundamental as they often have other concerns which make it difficult to take in new messages about early child development.

*"We forget the caregiver's wellbeing, (. . .) TAHEA comes—they want to train in nutrition or child safety and so on, (. . .) sometimes they* (the caregivers) *don't listen because they have other problems in mind"*

(Professional 6).

Professionals suggested ways to promote psychosocial stimulation, even in busy families, by integrating it into everyday activities. They gave examples on how to be creative in making toys and stimulate children without need for money.

*"When the mother performs her duties, does not leave the child, and when she is doing laundry, she tells her child that 'my child, this is how we do laundry', so the child knows that when I do this, it is called washing"*

(Professional 9).

*"If it is a ball, we look for rags or old clothes, make a ball and tie it properly with rope (. . .). If it is things that make sounds, we will pick gravel, rocks, they will use the rocks to hit on the soda caps, it makes sounds, and a child can play with it"*

(Professional 8).

**Issues of security and safety.** >A young child's development is vulnerable to psychological as well as physical stress. In our interviews, there was limited discussion on the topic of physical security and safety among caregivers. However, one of the professionals exemplified how stress can impair development and precautions should be taken to ensure the safety of the child.

*"Another dangerous factor is what we call violent homes, here I mean where at home there is no peace, the parents are fighting and quarrelling and even other relatives, it's very dangerous for the child's development"*

(Professional 5).

*"You have to observe his safety. There should be playing materials, so you have to ensure there shouldn't be dangerous objects around. You make sure that environment is safe for a child to learn by himself playing"*

(Professional 5).

## Suggestions and recommendations for psychosocial stimulation in SAM management

The barriers and facilitators identified in our interviews with caregivers and professionals include issues that can be addressed by taking a holistic view to creating a psychosocial stimulation programme for children treated for SAM, while other issues are more structural in nature and require broader societal initiatives. Based on the findings presented above, we have identified issues at the family-level as this is the relevant entry point for our intervention among caregivers of children with SAM. Here, we summarize the suggestions and recommendations for the design of a context-specific psychosocial stimulation package that can be integrated with the management of SAM in children (Fig 1).

## Discussion

This study explored factors to be considered when creating a relevant psychosocial stimulation programme for children treated for SAM in a Tanzanian context. Our results demonstrate both family-level and broader structural factors that caregivers and professionals believe may influence good nurturing care and which need to be reflected in a programme that could be helpful to families. The findings highlight a number of issues which may inhibit adoption of

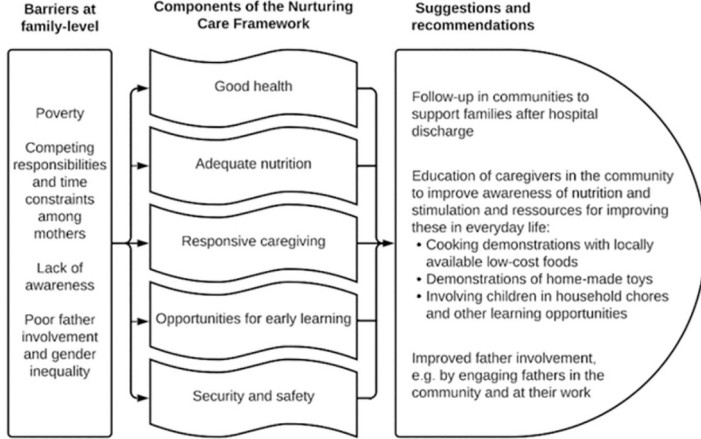

**Fig 1. Overview of identified barriers within the family, that are linked with the components of the Nurturing Care Framework, and suggestions and recommendations on how to support nurturing care and subsequent learning and development.**

the five components of the Nurturing Care Framework if not considered. These include financial and food insecurity, competing time constraints of caregivers, lack of awareness of the importance of responsive caregiving, play and stimulating activities, gender inequality and lack of father involvement, and finally, issues of security and safety of children.

Our interviews emphasise how caregivers and professionals are already cognisant of the components: good health and nutrition, required to meet a child's developmental potential, but barriers within society or their economy make it difficult for families to support children [47]. Professionals also emphasised the importance of caregiver wellbeing to enable them to provide supportive care for their children. On the other hand, caregivers we interviewed focussed less on the other components: responsive caregiving and opportunities for early learning, which seemed to be less familiar ways of improving children's development. Another component that was not highlighted much in interviews, was the importance of security and safety. This may need emphasising in counselling on psychosocial stimulation support for improving children's developmental outcomes—however we did not probe specifically into this area and how best to emphasise this might need more consideration.

In this study, we found that poverty and financial insecurity were commonly identified as barriers to provide key components of nurturing care; including adequate nutrition and opportunities for early learning. Mothers are generally knowledgeable on what constitutes good nutritional support for children, but many families have limited resources to act on this knowledge. Some professionals interviewed suggested that cooking demonstrations might improve understanding of how to provide better nutrition at a low cost from available ingredients. Professionals also suggested that educating mothers in low- and no-cost strategies for creating opportunities for early learning, would be possible. This includes encouraging families to make home-made toys from available materials and engaging children in household activities as a form of play. These suggestions were found relevant to our context in Tanzania and are likely to be generalizable broadly across low-income settings [48, 49]. However, the evident impact of poverty, particularly among families who bring their children into a nutrition unit, underlines a need for additional support through cash transfer programmes, social welfare systems, or similar more structural approaches–some of which have demonstrated evidence of improving child development [18, 50]. It is widely documented that poverty has a multi-faceted and detrimental effect on child development [51, 52] and we acknowledge the fundamental need for these broader issues to be addressed through structural changes, although these lie beyond the scope of our present programme.

Gender norms in the study setting may mean that women are responsible for household duties including raising children and men expected to be key earners [53]. Our interviewees projected responsibility for responsive caregiving on mothers and rarely mentioned other people within the circle of care. Many mothers who we interviewed described feelings of being overwhelmed and lacking time for additional activities. This may be common across the globe. Analysis of the UNICEF multiple indicator cluster surveys found that, across 38 low-income countries, 48% of fathers did not engage in any stimulation activities with their child [54]. Furthermore, a recent meta-analysis of 102 interventions to improve early child development outcomes found only 7% involved fathers in the intervention and only one study assessed paternal stimulation outcomes directly [31]. To shift this, it is likely that programmes would need to address cultural expectations of a father's role as well as the wider community's (including men's) understanding of the importance of responsive caregiving for children's brain development. It is our recommendation that this be considered in future programmes within community settings.

Many mothers of children suffering from acute malnutrition in the Mwanza region have no or low level of education [41]. Studies have demonstrated that maternal schooling is strongly

associated with nutritional outcomes of their children and that maternal knowledge regarding causes of malnutrition is a strong predictive factor of childhood stunting [41, 55]. It is widely reported that maternal schooling has a strong positive effect on child development [56]. Whilst the majority of these studies are from high income settings, a recent study in Uganda found that for each additional year of maternal education, the child's developmental index score rose by 0.12 increments, which translated into effects on ability to read, identify letters and numbers at age 3–4 years [56]. This suggests that not only educational interventions to improve maternal knowledge of psychosocial stimulation and nutrition could prove effective for children suffering from SAM in rural Tanzania, but more so, supporting girls to remain in school, might also be fundamental. Clearly, approaches to improve early child development should holistically encompass the needs of families, healthcare staff and communities and consider the wider structural and social determinants of health. This is complex and will require working with ministries and agencies who may be able to support this on a wider scale. Although these aspects are not addressed further within our specific programme in the nutrition unit, we hope that by continuing to highlight them, we can advocate for programmes which start early with support for young girls and women to stay in education.

Suggestions from caregivers and professionals on how to overcome some of the barriers highlighted included recommendations that could be implemented into a short, focussed intervention during management of SAM. These included better linkage with community health workers on discharge from the nutrition unit and continued education of caregivers to improve knowledge of both nutrition and psychosocial stimulation. Professionals provided specific recommendations on ways to provide advice to caregivers which could be used in everyday scenarios, e.g. cooking demonstrations, homemade toys and encouraging learning activities when involving children in everyday chores. Whenever possible, getting fathers more involved in care as well as generally widening the circle of care may make a big difference to enabling children to grow and develop better.

Earlier studies suggest various intervention strategies and programme models, which can be used to deliver parenting interventions and achieve improved child development [31]. In particular, interventions provided early and more intensively (e.g. weekly) with supportive supervision may be most effective [57]. The majority (70%) of published effective parenting interventions last at least 12 months [31] although several meta-analyses have shown no significant correlations between programme duration and developmental outcomes [31, 58] or between more parent contacts and larger effect sizes [59].

### Strengths and limitations

The aim of this study was to inform the development of a psychosocial intervention delivered to families during management of their child's malnutrition. Consequently, we focused mainly on barriers and facilitators for supporting early child development at the family level, although we did allow for topics relating to other levels to come up during interviews. We acknowledge that this may be considered a study limitation as other barriers and facilitators might have emerged if our interview guide had focused on the child, health system or community levels. Similarly, we did not obtain much information on the topic of psychological stress among caregivers, which may be a consequence of not probing specifically into this topic during interviews. Finally, we are aware that the issues raised may be subject to some degree of social desirability bias, and the reported practices and perceptions may not necessarily correspond to actual behaviours.

The study also had several strengths, including that our findings derive directly from caregivers whose children have SAM and professionals who work with malnutrition and child

development in Mwanza, Tanzania. These may be of relevance for other low-income settings, especially in sub-Saharan African countries. We hope that this study may shed light on some issues that could be addressed in creating stronger support for children to thrive if provided with both nutritional and psychosocial stimulation support in a programme embedded with a clear understanding of the wider social and structural issues which affect the care of a child admitted to the nutrition unit. Our qualitative approach has allowed us to gain a nuanced perspective on barriers for learning and development in young children by gaining points of view from both caregivers and professionals which we can share and learn from. Additionally, inclusion of professionals working at different levels within the hospital and at different NGOs contributed to a broader variety of perspectives.

## Conclusions

Based on the barriers, facilitators and suggestions we identified, we found that a context-relevant psychosocial stimulation programme to improve child development among young children treated for SAM in Tanzania should include better linkage with community workers on discharge from the nutrition unit, education of caregivers to improve awareness of nutrition and psychosocial stimulation, and involvement of fathers to widen the circle of care and decrease the burden on mothers. Specific recommendations from professionals on counselling of caregivers which could be useful in everyday scenarios included cooking demonstrations, homemade toys and encouraging learning activities by involving children in everyday chores. Other issues identified were structural, highlighting the continued need for other types of programmes to target food and financial insecurity.

## Supporting information

**S1 Questionnaire.**
(DOCX)

## Acknowledgments

The authors would like to thank the interviewees who took time to participate in this study and share their views and thoughts with us. We would also like to acknowledge the healthcare staff at the Bugando Medical Centre nutrition unit, who helped sampling and identifying the caregivers for interviews, as well as the staff at the National Institute for Medical Research, who also participated in this study.

## Author Contributions

**Conceptualization:** Erica Sanga, George PrayGod, Suzanne Filteau, Jayne Webster, Melissa Gladstone, Mette F. Olsen.

**Data curation:** Erica Sanga, George PrayGod, Happiness Kunzi, Theresia Setebe.

**Formal analysis:** Cecilie L. Jensen, Erica Sanga, Heather Kitt, Jayne Webster, Melissa Gladstone, Mette F. Olsen.

**Methodology:** Cecilie L. Jensen, Erica Sanga, Mette F. Olsen.

**Project administration:** Erica Sanga, George PrayGod, Mette F. Olsen.

**Supervision:** Erica Sanga, George PrayGod, Melissa Gladstone, Mette F. Olsen.

**Validation:** Erica Sanga.

**Writing – original draft:** Cecilie L. Jensen.

**Writing – review & editing:** Erica Sanga, Heather Kitt, George PrayGod, Happiness Kunzi, Theresia Setebe, Suzanne Filteau, Jayne Webster, Melissa Gladstone, Mette F. Olsen.

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
