## [Decision Letter · Decision Letter 0]

23 Oct 2023

PONE-D-23-11475Developing a context-relevant psychosocial stimulation intervention to promote cognitive development of children with severe acute malnutrition in Mwanza, TanzaniaPLOS ONE

Dear Dr. Olsen,

Thank you for submitting your manuscript to PLOS ONE. After careful consideration, we feel that it has merit but does not fully meet PLOS ONE’s publication criteria as it currently stands. Therefore, we invite you to submit a revised version of the manuscript that addresses the points raised during the review process.

We look forward to receiving your revised manuscript.

Kind regards,

Engelbert A. Nonterah, MD, PhD

Academic Editor

PLOS ONE

Journal Requirements:

2. Please include a complete copy of PLOS’ questionnaire on inclusivity in global research in your revised manuscript. Our policy for research in this area aims to improve transparency in the reporting of research performed outside of researchers’ own country or community. The policy applies to researchers who have travelled to a different country to conduct research, research with Indigenous populations or their lands, and research on cultural artefacts. The questionnaire can also be requested at the journal’s discretion for any other submissions, even if these conditions are not met.  Please find more information on the policy and a link to download a blank copy of the questionnaire here: https://journals.plos.org/plosone/s/best-practices-in-research-reporting. Please upload a completed version of your questionnaire as Supporting Information when you resubmit your manuscript3. We note that you have indicated that data from this study are available upon request. PLOS only allows data to be available upon request if there are legal or ethical restrictions on sharing data publicly. For more information on unacceptable data access restrictions, please see http://journals.plos.org/plosone/s/data-availability#loc-unacceptable-data-access-restrictions.  

Reviewers' comments:

Reviewer's Responses to Questions

**Comments to the Author**

1. Is the manuscript technically sound, and do the data support the conclusions?

Reviewer #1: No

Reviewer #2: Yes

2. Has the statistical analysis been performed appropriately and rigorously? 

Reviewer #1: N/A

Reviewer #2: Yes

3. Have the authors made all data underlying the findings in their manuscript fully available?

Reviewer #1: No

Reviewer #2: Yes

4. Is the manuscript presented in an intelligible fashion and written in standard English?

Reviewer #1: Yes

Reviewer #2: Yes

5. Review Comments to the Author

Reviewer #1: The topic of this study is highly significant at both local and global levels. Investment in promoting early childhood development can yield substantial health, social, and economic benefits for society as a whole. There is a dearth of psychosocial stimulation programs designed for children facing adversities in resource-constrained settings. However, the study could have several flaws, leading to limited results. No results on the barriers and facilitators at the child, community, healthcare providers, and the healthcare system levels were reported.

Below are my suggestions:

1. Avoid excessive use of acronyms.

2. Introduction: Provide a comprehensive explanation of psychosocial stimulation, both in general and specific to Severe Acute Malnutrition (SAM). Detail the WHO guidelines for inpatient SAM treatment and recommendations for psychosocial stimulation.

3. Methods: Provide a more detailed description of the design of the qualitative formative research in the "Study Design and Setting" section. Explain the rationale behind the chosen qualitative methods and the theoretical framework guiding the study.

4. Clarify whether the psychosocial interventions studied were intended for inpatient treatment in hospitals or community management of SAM.

5. Describe the process of selecting professionals for the study.

6. Specify the number of participants who declined to participate and their reasons for refusal.

7. Elaborate on the process of deciding to stop conducting interviews based on data saturation. Explain who made this decision and how.

8. Detail the development and pilot testing of the interview guides.

9. Address the limitations of the study's results, particularly their focus on barriers and facilitators at the family level.

Reviewer #2: This is a timely and insightful research related to child development and malnutrition particularly in low and middle income countries. It unearths key findings that are useful for interventions in similar context. It also provides nuanced information given the paucity of qualitative studies on the subject matter. While this is a well written manuscript, some aspects of the paper require revision.

The introductory information that builds up argument for the study appear limited. For instance, what is severe acute malnutrition (SAM)?, What does psychosocial stimulation entail and/or what are its components?. The introduction section can improve with a bit more information to better ground the study.

The study was guided by the nurturing care framework (NCF) by the WHO. Data was organized and synthesized using this framework. However, the framework has only been summarized without being thoroughly reviewed. For instance, what are its strengths or otherwise?, why was it the best option for this study and not some other framework out there?

Then again, what do the five components of the NCF mean. When you refer to ‘good health’ or ‘adequate nutrition’ etc , what does this mean for your study. In effect, if the framework was thoroughly reviewed, it will clear all doubts and help one see if findings reflect what you set out to do.

You state how caregivers were sampled (purposively). How were professionals sampled? How did you decide which professional to interview or were all professionals within the context of interest interviewed?

Under the results section, on page 10 you provide a quote from a professional on the wellbeing of the caretaker. Linking this to the comment above about how dearth the explanation of the NCF framework is , it becomes unclear exactly who the focus of the component of ‘good health’ is for. If the components are well explained, then it will be easier to link or see the barriers to promoting PS for children in SAM.

On the component of ‘security and safety’ on page 15, it is stated that there was limited discussion on psychological stress. Can a brief reason be given since the study’s focus is on PS?

6. PLOS authors have the option to publish the peer review history of their article (what does this mean?). If published, this will include your full peer review and any attached files.

Reviewer #1: No

Reviewer #2: No

---

## [Author Response · Author response to Decision Letter 0]

16 Jan 2024

Response to Reviewers

Below we provide a point-by-point response to the comments made by reviewers. We have revised our manuscript based on comments and suggestions and would like to acknowledge that these inputs have improved our manuscript. The line numbers refer til the clean version of the revised manuscript (file labeled “Manuscript“).

Reviewers' comments:

Reviewer #1: 

The topic of this study is highly significant at both local and global levels. Investment in promoting early childhood development can yield substantial health, social, and economic benefits for society as a whole. There is a dearth of psychosocial stimulation programs designed for children facing adversities in resource-constrained settings. However, the study could have several flaws, leading to limited results. No results on the barriers and facilitators at the child, community, healthcare providers, and the healthcare system levels were reported.

Below are my suggestions:

1. Avoid excessive use of acronyms.

Response: In the revised manuscript, we have reduced the use of acronyms.

2. Introduction: Provide a comprehensive explanation of psychosocial stimulation, both in general and specific to Severe Acute Malnutrition (SAM). Detail the WHO guidelines for inpatient SAM treatment and recommendations for psychosocial stimulation.

Response: We acknowledge that information was lacking on these topics and have thus made major revisions to the introduction section to include details about severe acute malnutrition, psychosocial stimulation and the relevance of stimulation interventions for children after exposure to SAM (line 59-107).

3. Methods: Provide a more detailed description of the design of the qualitative formative research in the "Study Design and Setting" section. Explain the rationale behind the chosen qualitative methods and the theoretical framework guiding the study.

Response: We have added a detailed description to the suggested section of the revised version (line 125-147).

4. Clarify whether the psychosocial interventions studied were intended for inpatient treatment in hospitals or community management of SAM.

Response: Our intention was to develop a psychosocial intervention that can be integrated with management of SAM both when delivered in hospitals and in communities. We have clarified this in the revised version (line 118-119).

5. Describe the process of selecting professionals for the study.

Response: The professionals (pediatrician, nutritionist, nurses in malnutrition ward, and key staff from TAHEA and EFFECTS) were also purposively selected based on their experience in working/ involvement with SAM children or projects related to malnutrition and/or early child development. We have added a description to the revised manuscript (line 169-173)

6. Specify the number of participants who declined to participate and their reasons for refusal.

Response: We have added these details to the first paragraph of the results section (line 250-253).

7. Elaborate on the process of deciding to stop conducting interviews based on data saturation. Explain who made this decision and how.

Response: The senior social scientist (E.S.) made this decision based on ongoing preliminary analysis of the interviews. We have added a more detailed description to the revised manuscript (line 193-197).

8. Detail the development and pilot testing of the interview guides.

Response: We have revised and re-organised the methods section to improve the description of the development and testing of the interview guides, which was previously spread over several paragraphs. The revised version now includes more detail on these processes (line 175-187).

9. Address the limitations of the study's results, particularly their focus on barriers and facilitators at the family level.

Response: Our study focused mainly on barriers and facilitators for supporting early child development at the family level, as the study was intended to inform development of an intervention delivered to families while in contact with health care teams during management of severe acute malnutrition. In addition, our topic guide focused on the views of caregivers and thus may have been biased to look more within the family. Although the focus of our results was on the family situation, our results do include barriers and facilitators at other levels as these emerged inductively when we coded the data, e.g. relating to ‘illness’ and ‘malnutrition’ at the child level, and we mention the need for links between nutrition units at the healthcare system level and community workers. In the revised manuscript, we now explain this as part of the methods section (line 178-182) and we’ve included it as a discussion point in the study limitations (line 526-536) 

Reviewer #2: 

This is a timely and insightful research related to child development and malnutrition particularly in low and middle income countries. It unearths key findings that are useful for interventions in similar context. It also provides nuanced information given the paucity of qualitative studies on the subject matter. While this is a well written manuscript, some aspects of the paper require revision.

The introductory information that builds up argument for the study appear limited. For instance, what is severe acute malnutrition (SAM)?, What does psychosocial stimulation entail and/or what are its components?. The introduction section can improve with a bit more information to better ground the study.

Response: We agree that the introduction could be improved. This was also commented by reviewer 1 and we have therefore rewritten the introduction section to improve the explanations of SAM, psychosocial stimulation and the relevance of providing stimulation to children exposed to SAM (line 59-107)

The study was guided by the nurturing care framework (NCF) by the WHO. Data was organized and synthesized using this framework. However, the framework has only been summarized without being thoroughly reviewed. For instance, what are its strengths or otherwise?, why was it the best option for this study and not some other framework out there?

Then again, what do the five components of the NCF mean. When you refer to ‘good health’ or ‘adequate nutrition’ etc , what does this mean for your study. In effect, if the framework was thoroughly reviewed, it will clear all doubts and help one see if findings reflect what you set out to do.

Response: We have provided more details about the NCF and our motivation for choosing this framework in the revised version (line 140-147).

You state how caregivers were sampled (purposively). How were professionals sampled? How did you decide which professional to interview or were all professionals within the context of interest interviewed?

Response: We acknowledge that this information was missing from the previous version, as also noticed by reviewer 1, and have now added the details to the method section (line 169-173).

Under the results section, on page 10 you provide a quote from a professional on the wellbeing of the caretaker. Linking this to the comment above about how dearth the explanation of the NCF framework is, it becomes unclear exactly who the focus of the component of ‘good health’ is for. If the components are well explained, then it will be easier to link or see the barriers to promoting PS for children in SAM.

Response: The comment relates to a quote about the importance of considering the health of caregivers as their (mental) health is central for them being able to take on messages about ECD and provide responsive care for their children. We agree with the reviewer that it could be argued that this does not belong with the “health” component of NCF which is mainly about the child’s health. In our revision, we have moved this part to the component on “awareness and prioritization of play and opportunities for early learning” as we believe its main message has to do with competing concerns and prioritization among caregivers (line 389-395).

On the component of ‘security and safety’ on page 15, it is stated that there was limited discussion on psychological stress. Can a brief reason be given since the study’s focus is on PS?

Response: This comment refers to a statement that was intended to say that there was limited discussion on the physical aspects of ECD (safety and security). We acknowledge that the statement was unclear and could be understood as limited discussion on the psychological stress. We have revised to clarify this (line 408-411). However, we also acknowledge that there may have been more discussion on physical aspects if we had probed specifically into this during interviews. We have added this as a limitation in the discussion section (line 526-536).

---

## [Decision Letter · Decision Letter 1]

26 Mar 2024

PONE-D-23-11475R1Developing a context-relevant psychosocial stimulation intervention to promote cognitive development of children with severe acute malnutrition in Mwanza, TanzaniaPLOS ONE

Dear Dr. Olsen,

Thank you for submitting your manuscript to PLOS ONE. After careful consideration, we feel that it has merit but does not fully meet PLOS ONE’s publication criteria as it currently stands. Therefore, we invite you to submit a revised version of the manuscript that addresses the points raised during the review process.

**Kindly review your manuscript based on the comments of the reviewer and return your manuscript within two weeks to facilitate further processing of same.**

We look forward to receiving your revised manuscript.

Kind regards,

Engelbert A. Nonterah, MD, PhD

Academic Editor

PLOS ONE

Journal Requirements:

Reviewers' comments:

Reviewer's Responses to Questions

**Comments to the Author**

1. If the authors have adequately addressed your comments raised in a previous round of review and you feel that this manuscript is now acceptable for publication, you may indicate that here to bypass the “Comments to the Author” section, enter your conflict of interest statement in the “Confidential to Editor” section, and submit your "Accept" recommendation.

Reviewer #2: All comments have been addressed

Reviewer #3: (No Response)

2. Is the manuscript technically sound, and do the data support the conclusions?

Reviewer #2: Yes

Reviewer #3: Yes

3. Has the statistical analysis been performed appropriately and rigorously? 

Reviewer #2: N/A

Reviewer #3: Yes

4. Have the authors made all data underlying the findings in their manuscript fully available?

Reviewer #2: Yes

Reviewer #3: No

5. Is the manuscript presented in an intelligible fashion and written in standard English?

Reviewer #2: Yes

Reviewer #3: Yes

6. Review Comments to the Author

Reviewer #2: The paper makes a valuable contribution to global health research and provides nuanced insights into using psychosocial stimulations to improve the well being and health outcomes of children in poor resourced countries.

The introduction of the paper has been restructured making it clearer and concise. There is less use of acronyms and this has boosted clarity.

The methods used have now been clearly explained and details the framework (NCF) that was used to analyze the data. Also good to see some reflexivity on the part of authors and research participants, as this allows for transparency.

There is also an insightful engagement with the findings in addition to providing contextual information removing any ambiguities. An outline of limitations has made the paper more focused. Bar one or two grammatical omissions, the paper is robust enough to be published.

Reviewer #3: Manuscript title: Developing a context-relevant psychosocial stimulation intervention to promote cognitive development of children with severe acute malnutrition in Mwanza, Tanzania

COMMENTS

I read the manuscript with keen interest. Overall, it is of relevance and well written. It also followed a scientific structure. Data management and analysis was done appropriately. Verbatim translation of recordings was done and this would ensure originality of contents. The coding process as described was thorough and the application of thematic content analysis was appropriate. A reliable tool (Dedoose) was used for the data processing/analysis. However, the authors’ statement on data availability gives an indication that access to data to supplement the manuscript is highly restrictive.

The framework used has to be illustrated clearly to relate well with the results and discussions. The results should be clearly presented to reflect the framework. The discussions should also reflect same. For instance, it was surprising that the authors mentioned “social”, “environmental” and “familial" factors in their discussion. These were not indicated anywhere in the results and methods section where the framework was mentioned.

Below are some other comments for consideration.

Study design and setting

• A brief description of the ‘BrightSAM (Brain development, growth and health in children with SAM) trial development study’ would help. Otherwise, it could be cited that this has been described elsewhere (e.g. reports, protocol, etc.).

• Sentence construction in line 135: It should be “exploring the views and experiences of a range "of" caregivers. The second "of" is missing.

• The framework used (Nurturing Care Framework) should be elaborated further to illustrate how the barriers and facilitators interact with the five components (good health, adequate nutrition, responsive caregiving, opportunities for early learning, and security and safety). Also, I suggest the framework is given a section/sub-section with a clear heading to make it more visible in the manuscript.

• In terms of flow, the description of the study setting as in line 149-160 could be moved to the beginning of the sub-section (study design and setting).

Participant selection and data collection

• The authors mentioned that “sampling was done with assistance from healthcare staff from the nutrition unit at BMC”. It would be clearer if this is elaborated a little further. How did the health staff help? What role did they play? Did they play a role in recruitment of the participants? If they played a role in recruitment, this could have influenced caregivers’ responses to questions relating to the health facility and caregiver. In that case, using the healthcare staff to recruit participants could introduce response biases which could be a limitation of the study. Such a limitation is worth mentioning in the manuscript.

• The number of interviews by category should be indicated. Also, an explanation of how such numbers were arrived at would be useful.

Results

• Is it possible to know how many participants were from the NGO and how many were healthcare workers? This should have also been addressed in the participant selection section.

• From the way the results are presented, the perspectives on suggestions and recommendations do not clearly standout. They are mixed with issues that came up in the perspectives on barriers and facilitators making it difficult for a reader to follow. Perhaps a separate heading could be created for suggestions and recommendations.

• In the summary (as indicated in the framework), it is not clear whether all the recommendations were perspectives of the study participants or they were based on the authors’ objectivity based on their analysis.

Discussions

• Line 436-468: the authors said “Our results demonstrate social, environmental and familial factors….”. From the results section, which of the factors are categorized as social, environmental and familial factors? This should be made clear either in the results section or in the discussion.

7. PLOS authors have the option to publish the peer review history of their article (what does this mean?). If published, this will include your full peer review and any attached files.

Reviewer #2: No

Reviewer #3: **Yes: **Aaron Kampim

---

## [Author Response · Author response to Decision Letter 1]

7 Apr 2024

Response to Reviewers’ Comments

Below we provide a point-by-point response to the second round of comments made by reviewers. We have revised our manuscript further based on these additional comments and suggestions and would like to acknowledge that these inputs have improved our manuscript. The line numbers refer til the clean version of the revised manuscript (file labeled “Manuscript“).

Reviewers' comments:

Reviewer #2: 

The paper makes a valuable contribution to global health research and provides nuanced insights into using psychosocial stimulations to improve the well being and health outcomes of children in poor resourced countries.

The introduction of the paper has been restructured making it clearer and concise. There is less use of acronyms and this has boosted clarity.

The methods used have now been clearly explained and details the framework (NCF) that was used to analyze the data. Also good to see some reflexivity on the part of authors and research participants, as this allows for transparency.

There is also an insightful engagement with the findings in addition to providing contextual information removing any ambiguities. An outline of limitations has made the paper more focused. Bar one or two grammatical omissions, the paper is robust enough to be published.

Response: We thank the reviewer for following up on the comments previously made. We are glad to learn that the reviewer found our last round of revisions useful, and we note that the reviewer has no further comments. 

Reviewer #3: 

I read the manuscript with keen interest. Overall, it is of relevance and well written. It also followed a scientific structure. Data management and analysis was done appropriately. Verbatim translation of recordings was done and this would ensure originality of contents. The coding process as described was thorough and the application of thematic content analysis was appropriate. A reliable tool (Dedoose) was used for the data processing/analysis. However, the authors’ statement on data availability gives an indication that access to data to supplement the manuscript is highly restrictive. The framework used has to be illustrated clearly to relate well with the results and discussions. The results should be clearly presented to reflect the framework. The discussions should also reflect same. For instance, it was surprising that the authors mentioned “social”, “environmental” and “familial" factors in their discussion. These were not indicated anywhere in the results and methods section where the framework was mentioned.

Response: We thank the reviewer for the positive feedback. We also acknowledge that access to the data from this study is highly restrictive under the current data protection regulations in Tanzania. However, data can be made available to other researchers upon reasonable request. We have given the relevant contact info in our ‘data availability statement’. Lastly, we agree with the reviewer that the Nurturing Care Framework could be further elaborated in relation to our presentation of results and discussion. We describe how we have done this in the point-by-point responses below.

Study design and setting

• A brief description of the ‘BrightSAM (Brain development, growth and health in children with SAM) trial development study’ would help. Otherwise, it could be cited that this has been described elsewhere (e.g. reports, protocol, etc.).

Response: A brief description is given in the first paragraph of the Methods section. We have now added a reference to a recent publication which included additional details about the study (Mwita et al., 2024). (Line 113-114)

• Sentence construction in line 135: It should be “exploring the views and experiences of a range "of" caregivers. The second "of" is missing.

Response: Thank you. The typo has now been corrected. (Line 145)

• The framework used (Nurturing Care Framework) should be elaborated further to illustrate how the barriers and facilitators interact with the five components (good health, adequate nutrition, responsive caregiving, opportunities for early learning, and security and safety). Also, I suggest the framework is given a section/sub-section with a clear heading to make it more visible in the manuscript.

Response: We have added a sub-heading to the paragraph describing the theoretical framework used. We aimed to explore facilitators and barriers in promoting early child development among children treated for SAM. We applied the structure of the framework with its 5 components to organize and interpret our findings, as we found that the expressed barriers and facilitators were each linked with one or more of the components. We have clarified this in the revised version (Line 150-160)

• In terms of flow, the description of the study setting as in line 149-160 could be moved to the beginning of the sub-section (study design and setting).

Response: We agree and have restructured the Methods section accordingly (Line 122-133)

Participant selection and data collection

• The authors mentioned that “sampling was done with assistance from healthcare staff from the nutrition unit at BMC”. It would be clearer if this is elaborated a little further. How did the health staff help? What role did they play? Did they play a role in recruitment of the participants? If they played a role in recruitment, this could have influenced caregivers’ responses to questions relating to the health facility and caregiver. In that case, using the healthcare staff to recruit participants could introduce response biases which could be a limitation of the study. Such a limitation is worth mentioning in the manuscript.

Response: We agree that involvement of healthcare providers could bias the responses given in a qualitative study about health services. However, in the present study, the health staff only helped the researchers to identify mothers with children who had recently been treated for SAM. They gave us their phone numbers and played no further role in the study. All communication about the study was with our study staff who had not been involved in the treatment of the child. We have clarified this in the revised version. (Line 165-168)

• The number of interviews by category should be indicated. Also, an explanation of how such numbers were arrived at would be useful.

Response: We’ve added the number of interviews by category of professionals to the Methods section (Line 179-182). In the previous version, we explained that the number of interviews was based on the principle of data saturation, but this only applies to the interviews with caregivers. The number of interviews with professionals was based on an ambition to include all categories of staff with relevant experience of treatment of severe acute malnutrition and/or promotion of child development in the setting. This has now been clarified in the revised manuscript (Line 177-179). We’ve also rearranged the text to give the information about numbers of interviews for both groups (caregivers and professionals) in the first paragraphs of ‘Participant selection and data collection’ (Line 164-184).

Results

• Is it possible to know how many participants were from the NGO and how many were healthcare workers? This should have also been addressed in the participant selection section.

Response: As described above, we’ve added the number of interviews by category to the method section as this was pre-defined by the availability of professionals with relevant experience. We refer back to these numbers in the result section to avoid repeating the information (Line 256).

• From the way the results are presented, the perspectives on suggestions and recommendations do not clearly standout. They are mixed with issues that came up in the perspectives on barriers and facilitators making it difficult for a reader to follow. Perhaps a separate heading could be created for suggestions and recommendations.

Response: We’ve structured the findings according to the 5 components of the NCF framework and we present these in a subsection called ‘Barriers and facilitators for providing psychosocial stimulation’. We agree with the reviewer that this subsection also presents suggestions and recommendations as these are closely linked with the barriers/facilitators. In line with the reviewer’s suggestion, we have now revised the following subsection called ‘Suggestions and recommendations for psychosocial stimulation in SAM management’ in which we sum up the relevant findings that can inform the design of a context-specific psychosocial stimulation package to be integrated with SAM management (our second aim of the study) (Line 430-442) 

• In the summary (as indicated in the framework), it is not clear whether all the recommendations were perspectives of the study participants or they were based on the authors’ objectivity based on their analysis.

Response: This section has now been revised to give the suggestions and recommendations for developing a PS package while more clearly indicating how these are supported by the study’s findings (line 430-442) 

Discussions

• Line 436-468: the authors said “Our results demonstrate social, environmental and familial factors….”. From the results section, which of the factors are categorized as social, environmental and familial factors? This should be made clear either in the results section or in the discussion.

Response: This was meant as a general characterization of our findings rather than a formal categorization. We have now revised to avoid introducing new terms in the description of findings (Line 446-447). 

Reference

Mwita, F.C. et al. (2024) ‘Developmental and Nutritional Changes in Children with Severe Acute Malnutrition Provided with n-3 Fatty Acids Improved Ready-to-Use Therapeutic Food and Psychosocial Support: A Pilot Study in Tanzania’, Nutrients, 16(5), p. 692. Available at: https://doi.org/10.3390/nu16050692.

---

## [Editor Report · Decision Letter 2]

16 Apr 2024

Developing a context-relevant psychosocial stimulation intervention to promote cognitive development of children with severe acute malnutrition in Mwanza, Tanzania

PONE-D-23-11475R2

Dear Dr. Mette F Olsen,

We’re pleased to inform you that your manuscript has been judged scientifically suitable for publication and will be formally accepted for publication once it meets all outstanding technical requirements.

Kind regards,

Engelbert A. Nonterah, MD, PhD

Academic Editor

PLOS ONE